# Regulation of Male Fertility by the Renin-Angiotensin System

**DOI:** 10.3390/ijms21217943

**Published:** 2020-10-26

**Authors:** Marta Gianzo, Nerea Subirán

**Affiliations:** 1Department of Physiology, Faculty of Medicine and Nursery, University of the Basque Country (UPV/EHU), 48940 Leioa, Spain; marta.gianzo@ehu.eus; 2Innovation in Assisted Reproduction Group, Biocruces-Bizkaia Health Research Institute, 48903 Barakaldo, Spain; 3Research and Development Department, MEPRO Medical Reproductive Solutions, 20009 San Sebastian, Spain

**Keywords:** Renin-angiotensin system, male infertility, angiotensin, renin, testis, spermatozoa

## Abstract

The renin-angiotensin system (RAS) is a peptidic system known mainly for its roles in the maintenance of blood pressure and electrolyte and fluid homeostasis. However, several tissues and cells have been described to possess an intrinsic RAS that acts locally through different paracrine and autocrine mechanisms. In the male reproductive system, several components of this system have been observed in various organs and tissues, such as the testes, spermatozoa and seminal fluid. Some functions attributed to this local RAS are maintenance of seminal plasma electrolytes, regulation of steroidogenesis and spermatogenesis, and sperm functions. However, their specific actions in these locations are not fully understood. Therefore, a deep knowledge of the functions of the RAS at both the testicular and seminal levels could clarify its roles in male infertility and sperm physiology, and the different RAS elements could be used to design tools enabling the diagnosis and/or treatment of male infertility.

## 1. Introduction

According to the World Health Organization (WHO), infertility is defined as the inability to achieve pregnancy after a year of normal sexual relationships without the use of any contraceptive method. Infertility is one of the most prevalent chronic health disorders involving young adults, caused by advanced parental age and our lifestyle among other factors [1]. Suffering one in six couples of reproductive age, this disease has a current estimated prevalence of 14% worldwide and it affects both men and women; in fact, 50% of cases are due to the female partner, and the other 50% are due to the male partner [1,2].

Male infertility is mainly caused by anatomical defects, genetic diseases and injuries, as well as testicular sperm and hormonal dysfunction [3,4,5]. The production of fully functional spermatozoa capable of movement and fertilization is under strict endocrine, paracrine and autocrine regulation. Deregulation of this hormonal control or alterations in the different key cellular communication systems can cause male infertility [4,5]. However, the precise causes of male infertility remain unexplained in approximately 30% of cases, as many reproductive defects cannot be detected with current diagnostic methods [2].

RAS is a communication system that is reported to play a key role in the regulation of reproductive function in both males and females [6,7]. In the female reproductive system, the RAS is involved in many physiological and physiopathological processes, such as oocyte maturation and quality control, endometrial lining production and/or hormone production, polycystic ovary syndrome, ovarian hyperstimulation syndrome and ovarian or endometrial cancer [7]. In this review, we focus on providing an update of recent important findings of the role of RAS in male reproductive function, which is capable of the regulation of male fertility at multiple levels [6,8,9]. In fact, multiple RAS family members are expressed on different male reproductive tissues, such as the testes and semen, where they regulate male fertility, acting synergistically with and/or independently of systemic RAS [6].

## 2. The Renin-Angiotensin System (RAS)

The RAS is a peptidic system with endocrine characteristics known mainly for its importance in the maintenance of blood pressure (BP) and electrolyte and fluid homeostasis [6]. This system is canonically considered to be a circulating hormonal system that exerts its functions through angiotensin II (Ang II) and aldosterone. This pathway is activated when renin is released by the kidneys into the bloodstream in response to decreased BP, sympathetic nervous system activation and/or sodium depletion [6]. This renin acts on angiotensinogen (AGT) of hepatic origin, forming angiotensin I (Ang I). The inactive Ang I peptide is hydrolysed by the angiotensin-converting enzyme (ACE), giving rise to the peptide with the highest activity in this system, Ang II, which ultimately exerts its action by interacting with Ang II type 1 and 2 receptors (AT1R and AT2R) [6,10]. Ang II signalling produces several responses, such as vasoconstriction of the peripheral circulation, increased sympathetic system activity, release of vasopressin, increased cardiac chronotropism, and release of aldosterone from the adrenal cortex, which can restore blood volume and pressure. In turn, Ang II itself inhibits the synthesis and release of renin, regulating the activation of the RAS [6].

This traditional concept of the RAS has been revised due to the discovery of new RAS family members, which has provided evidence of the existence of new non-canonical pathways of this system (Figure 1). Principally, the field was redefined by the discovery of *ACE2*, an ACE homologue that cleaves Ang II to generate angiotensin 1–7 (Ang 1–7), which activates the Mas receptor (*MasR*). Later, the list of pathway members was extended when another fragment of Ang II, Ang 3–8 (also called Ang IV), was observed; Ang 3–8 acts on its own receptor, insulin-regulated aminopeptidase (IRAP, also called AT4R) [11]. Finally, the (pro)renin receptor (PRR), on which prorenin and renin act directly, was described [12]. The appearance of all these new RAS components has implied that this response system involves complex interplay among various angiotensin receptors and their signalling pathways, as well as receptor-like activity of RAS enzymes that can promote Ang II-independent intracellular enzymatic pathways [11].

In recent years, attention has also been paid to evidence of a widespread local tissue RAS. This new concept emerged after discovery of the existence of different RAS family members in several tissues that are able to regulate several specific functions, working synergistically with or independently of the systemic RAS [6,11]. These local RASs are mainly characterized by 1) the presence of different RAS components, such as AGT and converting enzymes; 2) local synthesis of Ang II and other angiotensins; and 3) the presence of the specific receptors. However, the local production of bioactive peptides is not necessarily dependent on local expression of all components of the local tissue RAS, since components from the circulation, such as renin, can also be taken up [7]. In fact, it has been observed that local RASs are present in a wide range of systems, organs, and tissues, such as the kidneys, brain, cardiovascular system, pituitary gland, adipose tissue, skin, adrenal gland, and female and male reproductive systems; these RASs play important roles in many physiological processes, such as, cell growth, extracellular matrix formation, vascular proliferation, endothelial function and apoptosis [11,13]. Furthermore, local RASs can also interact with other signalling pathways, including those involving tachykinins, enkephalins, nitric oxide (NO), prostaglandins or cellular phosphatases [6,13].

Along with the local synthesis and uptake of RAS elements by tissues, there is increasing evidence that both synthesis and uptake also occur at the cellular level, which suggests a new mechanism of action for a physiological system called the intracellular RAS [14]. This new intracellular system is characterized by the presence of different RAS components inside the cell and by the synthesis of Ang II at an intracellular site. The concept of this system is based on 1) observation of the existence of diverse isoforms of AGT and renin (as a result of glycosylation and alternative splicing, respectively) and different forms of ACE (intracellular and secreted), 2) the existence of alternative enzymes for the synthesis of Ang II (such as cathepsin and chymase), and 3) intracellular detection of these components under particular cellular conditions. In addition, these components must be able to mediate biological effects from an intracellular location to be functionally relevant [15]. In fact, changes in cell structure and gene transcription have been reported to induce the mobilization of intracellular calcium deposits and stimulate the growth of different cell types have been reported [13,15]. In spite of that, the functional roles of the intracellular RAS in physiology and pathophysiology have not yet been fully elucidated. 

### Axes that Compose the RAS

Nowadays, RAS has been described as a complex system composed by the canonical axis as well as the non-canonical axes (Figure 1), whose components are widely expressed in all cell types and organs in humans. Therefore, RAS plays important roles in numerous physiological events, such as renal, neuronal, cardiac, pancreatic, vascular, adrenal, pituitary, cognitive, ageing-related, inflammatory and reproductive processes [16]. Notably, one of the most relevant functions of the RAS is its participation in various processes related to male fertility [7,8,17,18].

Renin/ACE/Ang II/AT1R/AT2R axis is a well-known pathway as it is considered the canonical or systematic pathway, which is mainly formed by AGT; renin; ACE; Ang I and Ang II and their receptors, AT1R and AT2R. Angiotensin III (angiotensin (2–8) or Ang III) is produced from Ang II through the action of the enzyme aminopeptidase A (APA) and exerts its actions through AT1R and AT2R [6,10]. Traditionally, the main functions of this axis were considered to be the maintenance of BP and electrolyte and fluid homeostasis. Other functions have since been attributed to the axis at the local and intracellular levels, which differ depending on the receptor that triggers the signal [6,15]. AT1R is definitively known to operate through various signalling mechanisms, such as by increasing the intracellular levels of Ca^2+^ (by increasing the influx of extracellular Ca^2+^ and the mobilization of intracellular Ca^2+^); activating various kinase pathways, including the mitogen-activated protein (MAP) kinase pathway; and activating the epidermal growth factor receptor (EGFR) in the plasma membrane. In contrast, the effects of AT2R stimulation are mediated mainly by phosphatases [19]. Therefore, the effects of AT2R- and AT1R-mediated signalling have been considered to be antagonistic [6,10,19]. AT1R has thus been associated with physiological responses such as vasoconstriction, the inflammatory response, cell proliferation or oxidative stress, while AT2R has been associated with processes such as vasodilation; apoptosis; and anti-inflammation, anti-cell proliferation and anti-oxidative stress processes [19].

The proteolysis cascade of *ACE2*/Ang(1–7)/*MasR* non-canonical axis starts with *ACE2*. This enzyme cleaves Ang II to directly generate Ang (1–7), which exerts its action by binding to *MasR*. Additionally, Ang I can be cleaved by many other peptidases, especially neutral endopeptidase (NEP), which can generate Ang (1–7). Finally, this heptapeptide can also be generated via the hydrolysis of Ang I by *ACE2* to form angiotensin (1–9) (Ang (1–9)), which is subsequently cleaved by NEP or ACE. However, it is important to highlight that this pathway seems to be catalytically less efficient than the ones mentioned above [20]. Different studies have observed antagonistic effects between the proteins Ang II and Ang (1–7) as well as among the receptors on which they act, the G-protein coupled receptors (GPCRs) *MasR* and AT1R [7]. Indeed, the beneficial effects of this axis encompass various biological processes, such as vasodilation and the stimulation of bradykinin and NO release [20].

The Ang IV/AT4R-IRAP axis is another non-canonical axis. In this signalling pathway, Ang III is transformed into angiotensin (3–8), also called angiotensin IV (Ang IV), by the action of aminopeptidase N (APN) and aminopeptidase B (APB). Ang IV ultimately acts through binding to AT4R/IRAP [21]. This axis plays key roles in the regulation of cognitive functions such as learning and memory, renal metabolism, cardiovascular damage, modulation of glucose uptake into cells, and regulation of the growth of several cell types [13].

Finally, the (pro)renin/PRR axis has been described after discovery of the existence of the renin receptor (PRR) [12], which acts independently of the classical axis after activation by renin itself and its precursor, (pro)renin [12,22] and it has been linked to cardiovascular, renal and degenerative diseases [22].

## 3. The RAS and Male Fertility 

The establishment of male fertility comprises a series of intricate and highly structured steps that depend on complex orchestration of communication systems, especially the RAS. In fact, numerous components of this system have been described in many organs and tissues of the male reproductive tract, including the epididymis [23,24], vas deferens [25], prostate [26], seminal fluid [27,28,29], testes [9,30], and spermatozoa [8,31,32,33]. In addition, accumulating data from in vitro, animal and clinical studies have indicated that this peptide system is involved in the correct functioning of the human male reproductive system and is frequently altered or deregulated in pathological conditions [8,18,22,29,31,32,33].

### 3.1. Regulation of Testicular Function by Local RAS

The main functions of the testes are the formation of spermatozoa, or spermatogenesis, and the production of testosterone, or steroidogenesis. Considering the importance and complexity of these processes, it is easy to understand the strict endocrine, paracrine and autocrine regulation to which they must be subject and how their deregulation can contribute to male infertility [4,5]. The first point of hormonal regulation of testicular function is the hypothalamic-pituitary-testicular axis [5]. Several members of the RAS are present in the testes, such as renin and Ang II, and are regulated by sex hormones as well as by gonadotropins [30,34] (Figure 2). Likewise, various paracrine and autocrine mechanisms modulate testicular function at the local level by acting on different types of cells present in this tissue, such as Sertoli, Leydig and/or spermatogenic cells [4]. At this point, the local RAS becomes involved; this RAS is isolated from the plasma RAS by a testicular blood barrier that protects male fertility from substances such as AT1R blockers and ACE inhibitors [22]. Within this local system, numerous researchers have found evidence of the synthesis and presence of components of different axes of the RAS.

#### 3.1.1. The Renin/ACE/Ang II/AT1R/AT2R Axis

All members of this axis have been found in the testes of various mammals, including humans (Table 1). The initial studies carried out in this regard showed that the major elements of this axis, AGT [34] and renin [35,36,37], are present in Leydig cells. At first it was thought that these proteins originated in the systemic RAS, since the testicular levels of AGT and renin and subsequently ACE, Ang II and its receptors increase at the beginning of puberty with the appearance of LH and FSH in the bloodstream [8,38,39]. Later, it was observed that their transcripts are synthesized in Leydig cells [6,34,40], indicating their local production and the linked regulation between the systemic and local RAS [22]. In addition, it has been found that increases in renin levels cause increases in testosterone synthesis [38]; however, further studies are necessary to elucidate whether such increases in testosterone are due to direct action on the PRR or signalling through the canonical pathway. Likewise, Ang I has also been found in the testes, mainly in Leydig cells [41], serving as a substrate for ACE. In fact, the testes also contain much higher concentrations of ACE (also known as ACE1) than other organs [6,42]. Interestingly, in the testes, two isoforms of this enzyme, somatic ACE (sACE) and testicular or germinal ACE (tACE or gACE), have been observed. Even though both are transcribed from the same gene, they differ structurally by the presence/absence of a 66-amino acid sequence [43] and consequently present different molecular structures. sACE is formed by two identical subunits, whereas tACE is formed by a single subunit [44,45,46,47] and both exhibit comparable enzymatic activity [6]. The transcription of the testicular form is tissue-specific and occurs as a result of alternative splicing, alternative transcription initiation, and alternative polyadenylation [9,48,49]. The most notable differences between the isoforms are their locations; while tACE is present only in male germ cells [9,50,51], being found in high concentrations during spermiogenesis [6,9], sACE is expressed in other testicular cells, such as Leydig cells and endothelial cells of the testicular interstitial tissue [9], and in a soluble form in seminal plasma [42]. Although the concentration of sACE in the testes is among the highest in all organs the specific role of sACE in the testes remains unknown. sACE-deficient mice are fertile and the testes might be the sources of this enzyme in seminal fluid [6], where it protects sperm during and after transfer to females [52].

The Ang II peptide is present in both germ cells [35] and in Leydig cells [41]. The presence of a blood-testicular barrier, together with the fact that all the components necessary for the production of this protein (renin, ACE, AGT and Ang I) are found within Leydig cells, supports the intracellular synthesis of this peptide. Ang II has been reported to be capable of inhibiting adenylate cyclase activity in rat Leydig cells, reducing basal and gonadotropin-stimulated cAMP and testosterone production [53,54]. On the other hand, it has also been proposed that Ang II may play a role in testicular growth and/or differentiation [55]. Finally, the Ang II receptors AT1R and AT2R have also been detected in rat, monkey and human testes, specifically in Leydig cells [53,56,57]. However, the protein levels of both receptors, as well as the levels of their respective mRNAs, depend on age; for example, the expression of AT2R predominates in the first days of life but gradually decreases until the fourth week of life, leaving AT1R to be almost exclusively expressed [40]. It is unknown whether these receptors play a role in the regulation of testicular formation [6]. Notably, two subtypes of AT1R, AT1AR and *AT1BR*, have been described in the testes of mice but not in those of humans, although no specific functions have been established for either of these subtypes [21]. The presence of AT1R has also been observed in rat and human seminiferous tubules, specifically in sperm cells at different maturation stages (spermatogonia and spermatids), suggesting that this receptor could be involved in spermatogenesis [57]. Finally, Ang II, through its binding to AT1R, may interfere with testosterone production [55].

Additional studies have shown that Ang II is transformed into Ang III by the action of APA, after which Ang III exerts its action by binding to AT1R and AT2R [6,10]. The APA enzyme has been detected in rat testis homogenates, showing high enzymatic activity. It has been proposed that this enzyme, like Ang II, is involved in inhibiting steroidogenesis through production of Ang III and activation of AT1R [58].

Taken together, these findings affirm that this canonical axis modulates steroidogenesis. Specifically, Ang II and III, through binding to AT1R, negatively regulate testosterone production. Moreover, AT1R may be involved in the spermatogenesis process. Although future studies are necessary, it has been suggested a positive regulation of steroidogenesis through the AT2R, considering the contrasting effects of AT1R and AT2R.

#### 3.1.2. The *ACE2*/Ang (1–7)/*MasR* Axis

All the principal components of the axis *ACE2*/Ang (1–7)/ *MasR* have been detected in rat, mouse, and human testes [32]. Single-cell RNA sequencing data on human testes showed predominant expression of *ACE2* in spermatogonia, Leydig and Sertoli cells [59], but *ACE2* has been found to be localized only in Leydig cells at the protein level [60,61]. Although *Ace2*-null mice are fertile [62], men with severely impaired spermatogenesis have lower levels of *ACE2* than fertile men, suggesting that this enzyme may modulate sperm formation [32]. *ACE2* has also been reported to play key roles in the regulation of testosterone production and in the local vascular regulatory system, in which it balances interstitial fluid volume by modulating the conversion of Ang II to Ang I [52]. Ang (1–7) has also been identified and characterized in mouse [61,63], rat [63] and human testes [32], in the cytoplasm of Leydig cells, and in Sertoli cells and primary spermatocytes at lower levels [32]. This heptapeptide is involved in the regulation of spermatogenesis, since lower levels of Ang-(1–7) have been found in men with severe spermatogenesis impairment than in fertile men [32]. Similar to this peptide, *MasR* has been described to be present in the cytoplasm of mouse [60,63], human Leydig cells and inside the human seminiferous tubules in all layers of the normal seminiferous epithelium, being equally distributed between interstitial and tubular compartments [32]. However, its mRNA has been detected in both Leydig and Sertoli cells, with its expression being more pronounced in the latter, but not in developing germ cells [60]. Concerning *MasR* function, *MasR*-deficient mice have constitutive alterations in the activity of genes encoding steroidogenic enzymes within the testes [60,64]. This finding and the observation that *MasR* and Ang-(1–7) are present in Leydig cells, suggest that both *MasR* and Ang-(1–7) could play a key role in modulating the production of testosterone [32]. Notably, it has been reported that *MasR*-deficient mice show marked reductions in testis weight, a significant increase of apoptotic cells during meiosis, the presence of giant cells and vacuoles in the seminiferous epithelium, and striking reductions in daily sperm production due to disturbed spermatogenesis [63,65], although the total numbers of Sertoli and Leydig cells are comparable in both wild-type and knockout animals [65]. Moreover, *MasR* levels are notably decreased or absent under conditions of severe spermatogenesis alteration in humans, providing insights into the role of the *MasR* in the regulation of spermatogenesis [32].

Although there are no data in the literature that specifically show the presence of Ang (1–9) in the testes, the presence of Ang I and Ang (1–7), as well as the enzymes ACE, *ACE2* and NEP, could indicate that this peptide is present in this tissue [6,7]. On the other hand, NEP has been detected in human testes [66] and in the membranes of rat Sertoli cells [67], and its activity is low in testicular homogenate [66] but high in Sertoli cells [67]. It has been suggested that this metalloendopeptidase may be related to sperm maturation and proacrosin activation [66]. Nevertheless, NEP-deficient mice have normal testicular function [68]. Similarly, an isoform of this enzyme, *NEP2*, has been observed in mouse [69] and human testes [70]. Studies have also demonstrated the presence of *NEP2* mRNA in the seminiferous tubules, specifically in developing germ cells and mainly in spermatids [71]. However, the concrete function of *NEP2* in the testes remains unknown.

Taken together, these data clearly indicate that this axis plays key roles in regulating steroidogenesis and spermatogenesis (Table 1).

#### 3.1.3. The Ang IV/AT4R-IRAP Axis

This pathway begins with the action of APN or APB. The metallopeptidase APN has been localized in both human [72] and mouse testes, showing high gene expression in Sertoli and Leydig cells [73]. APN seems to play a role in the inhibition of testosterone synthesis [74]. In fact, mice lacking APN present defects in spermatogenesis and infertility, as they have altered Sertoli cell function [73]. In addition, APB has been detected in mouse testes, and it also participates in inhibition of the synthesis of this hormone [74] (Table 1).

Although the specific function of Ang IV in the testes remains unknown, Ang IV could also be involved in inhibition of testicular testosterone production [58,74]. Likewise, although the presence of AT4R/IRAP in the testes has not yet been described, the presence of the other components of this axis could indicate that this receptor is present in this tissue. Despite the scarcity of evidence regarding the presence and functions of the main members of Ang IV/AT4R-IRAP axis in the testes, the available data show how this axis is also related to inhibition of testicular testosterone production and, therefore, in the regulation of spermatogenesis.

#### 3.1.4. The (Pro)renin/PRR Axis

Prorenin is known to be produced and secreted by the testes [75] and it has been found a correlation between sperm density and the prorenin level in semen [76]. Similarly, it has been shown that Leydig cells are capable of producing renin [6,34,40] and that renin levels are directly related to testicular testosterone concentrations [38]. Thus, it could be hypothesized that this axis is positively involved in steroidogenesis (Table 1). Nonetheless, as previously described, the (pro)renin/PRR axis was described after the existence of the PRR was discovered [12]. No evidence has yet been reported showing that this receptor is present in the testes; therefore, further studies should be performed in order to demonstrate the presence of PRR and consequently the existence of this axis as well as to understand the functional role of prorenin and/or renin independent of the effects of the classical RAS cascade.

### 3.2. Regulation of Sperm Phisiology by Local RAS 

In order to acquire fertilization ability, spermatozoa must undergo a series of processes, such as motility acquisition; capacitation; the acrosomal reaction; and oocyte recognition, fusion and activation [77]. These processes must be perfectly regulated to ensure proper development. It is known that tight control is exerted by the joint actions of complex biological systems, including the RAS [21,40,77]. Accumulated evidence regarding the presence, distributions and specific functions of different members of this peptide system in sperm cells suggests that the RAS regulates male reproductive function by acting directly on the fertility potential of spermatozoa [8]. In addition, as has been observed for testicular function, the different RAS axes could exert opposite roles, which would result in fine modulation of reproductive function (Table 2).

#### 3.2.1. The Renin/ACE/Ang II/AT1R/AT2R Axis

Members of the Renin/ACE/Ang II/AT1R/AT2R are present in seminal fluid and sperm cells, where they may have different putative effects. High levels of renin and Ang II are present in seminal plasma [27,78]; although they have not yet been specifically detected in sperm cells. sACE has also been observed in seminal plasma at higher concentrations than in blood plasma and it is positively correlated with sperm concentration and motility [40,76], although oligospermic patients have sACE levels similar to those of normozoospermic men [79]. Even though AGT has not been described in spermatozoa, it plays a key role in the function of these cells, since *Agt*-deficient male mice show reduced fertility, as their sperm cells are less likely to fertilize oocytes than those of wild-type mice [80].

On the other hand, other RAS members, such as tACE, Ang II, and Ang II receptors, have been specifically detected in germinal cells. Specifically, tACE is present in spermatids and mature sperm of different species [81,82,83], being located mainly in the acrosomal region, the equatorial segment, the postacrosomal region and the intermediate piece of sperm cells [84,85]. Although tACE enzymatic activity is low in immature rats, it increases with sexual maturity, suggesting that its presence is dependent on sexual maturation [51,86,87,88,89,90]. Moreover, different studies have demonstrated the involvement of tACE in sperm motility [90,91,92,93], capacitation [87,89], the acrosome reaction [89] and sperm-oocyte fusion [88]. Interestingly, the levels of tACE on the sperm surface differ during different phases of the fertilization process because tACE is released during capacitation [87,89] and the acrosome reaction [89]. It has been supposed that this release may have a physiological role in the regulation of sperm physiology as well as in sperm fertilization ability [88]. In this sense, the lack of sACE in male mice does not affect fertility, while animals lacking both ACE isoforms show reduced fertility due to failure of spermatozoa transport through the oviduct and gamete fusion [94,95]. On the other hand, tACE has been proposed to be an essential protein in the sperm-oocyte fusion process [96], although specific tACE3-IZUMO1 binding is not required for oocyte fertilization [52]. In addition, patients enrolled in in vitro fertilization (IVF) programmes found that aberrant, reduced or absent expression of the tACE protein on sperm underlies failed fertilization [51], and it is also involved in embryo development, since higher percentages of tACE-positive cells and fewer enzyme molecules per spermatozoon in semen samples are positively correlated with better embryo quality and development [93].

Ang II has been widely studied in sperm cells and both Ang II receptors, AT1R and AT2R, have been identified in sperm cells. Ang II is evidently a key modulator of sperm function given that this peptide is involved in sperm motility, capacitation, and the acrosomal reaction [88,89,97,98,99]. AT1R has been detected in developing spermatids and mature spermatozoa in humans and other mammalian species [33,98,99,100,101] and is located in the tails of human sperm [100]. Targeted deletion of the *AT1BR* gene (*Agtr1b*) in mice results in high *AT1BR* transcriptional activity in mature and immature sperm cells [6]. Treatment with antagonists of ATIR receptor (losartan) inhibits sperm motility and Ang I and II increases not only the percentage of motile spermatozoa but also the linear velocities of the spermatozoa, which proves that AT1R is involved in sperm motility [97]. AT2R has been described in mouse [98] and human spermatozoa [33], and it may play also an important role in sperm motility, as it has been observed that AT2R is related to human sperm concentration and motility, and the percentage of AT2R-positive spermatozoa is lower in asthenozoospermic patients (sperm samples with suboptimal motility) than in normozoospermic males [33].

In conclusion, this canonical axis plays a key role in the proper functioning of sperm cells since it is involved in physiological processes such as sperm motility, capacitation, the acrosome reaction, and sperm-egg recognition and fusion (Table 2).

#### 3.2.2. The *ACE2*/Ang (1–7)/*MasR* Axis

Few studies describe the presence and function of the *ACE2*/Ang (1–7)/*MasR* axis in spermatozoa. Although the presence of *ACE2* in sperm cells has not yet been demonstrated, several evidences indicate the existence and/or action of the other main members of this signalling pathway, such as Ang 1–7 and *MasR* (Table 2). Even though Ang (1–7) has not been localized in mature spermatozoa [32,102], incubation of seminal samples from asthenozoospermic patients with this peptide increases the proportion of progressive motile spermatozoa through *MasR* signalling [102]. In addition, in the sperm of asthenozoospermic patients, Ang-(1–7) may exert some effects independent of *MasR*, perhaps through binding to receptors such as the MRGPRD receptor (a GPCR related to *MasR*) or AT2R [102]. With reference to the specific receptor, it has been observed that *MasR* is present in human mature spermatozoa at the mRNA and protein levels [32,102]. *MasR* is located in the human sperm head, with the highest levels at the acrosomal region, but also in the tail. In addition, this receptor is functional, and its activation participates in regulating sperm motility [102].

The presence of another enzyme in this non-canonical RAS axis, NEP, has also been broadly proved. In fact, this endopeptidase has been located at higher concentrations and with stronger enzymatic activity in human seminal fluid than in other body tissues [28]. It has also been detected at the gene and protein levels in sperm cells; it is located in the neck in a small proportion of human spermatozoa [103]. Congruent with its location, this enzyme participates in the regulation of sperm motility, since its inhibition improves the motility of capacitated spermatozoa, increasing the percentage of sperm with progressive motility [103]. Moreover, NEP activity is altered in asthenozoospermic patients compared to normozoospermic men [29]. Likewise, the NEP-homologous enzyme *NEP2* [69] has been described to be present around the equatorial segment in human spermatozoa, and its selective inhibition with phosphoramidon increases sperm progressive motility [71]. However, the effects observed after inhibition of both neprilysins are mediated, at least in part, by tachykinins [71]. In addition, animal studies have shown that *NEP2*-deficient mice exhibit less efficient egg fertilization than wild-type mice and that large proportions of oocytes do not develop normally after fertilization with *NEP2-null* mouse spermatozoa [68].

#### 3.2.3. The Ang IV/AT4R-IRAP Axis

The all components of the Ang IV/AT4R-IRAP axis have not yet been detected in spermatozoa. In spite of that, presence of APN, the enzyme responsible for the formation of the Ang IV peptide, could indicate that both this peptide and its receptor are present in these cells. APN is present in human seminal fluid [28] and exhibits significantly higher activity and concentrations in seminal fluid than in other body tissues [28]. In sperm, this enzyme has also been detected at the gene and protein levels; it is located at the plasma membrane of the equatorial/postacrosomal region of the sperm head, in the neck and uniformly distributed along the tail [103]. In mussel spermatozoa, this enzyme induces the acrosome reaction [104,105]. Moreover, APN has been negatively related to sperm motility [31,103,106,107], and its activity is altered in asthenozoospermic patients compared to normozoospermic men [29] (Table 2). Moreover, the alteration in APN activity induces cellular toxicity in sperm cells, which causes persistent damage [29,106] and produces several adverse consequences and affecting early embryonic development in mice [106,107]. These findings are consistent with the fact that the cumulative probabilities of having developed blastocysts and viable embryos increase by 1.5-fold when semen samples with relatively low APN levels are used during the intracytoplasmic sperm injection (ICSI) technique (Gianzo et al., unpublished data). Strong evidences, therefore, indicate that APN play a key role in sperm fertility ability.

#### 3.2.4. The (pro)renin/PRR axis

Large amounts of renin and its precursor, prorenin, have been detected in human seminal fluid [76,78]. However, the putative effects of these molecules on the physiology of spermatozoa have not yet been clarified. As previously indicated, the presence of PRR is essential for the existence of the (pro)renin/PRR axis [12]. In addition, we have recently described the presence of PRR in human spermatozoa at the gene and protein levels and this receptor seems to play a role in sperm motility (Gianzo et al., unpublished data). PRR is located mainly in the proximal region over the acrosome and in the postacrosomal region of the head; to a lesser extent, it also extends along the sperm tail. Given these novel data, along with the high concentrations of renin and its precursor, prorenin, observed in human seminal fluid [76,78], it can be assumed that this non canonical axis could be important in the regulation of sperm functions (Table 2).

## 4. Conclusions and Perspectives 

The descriptions of different elements in the canonical and non-canonical enzymatic RAS pathways in the male reproductive system indicate that both pathways may be actively present, regulating male fertility by paracrine and autocrine mechanisms. In the testes, several RAS components are present in Leydig, Sertoli and spermatogenic cells at different maturation stages, indicating that these elements can modulate testosterone production and spermatogenesis. RAS functions at the seminal level are involved in the regulation of essential processes of male fertility, such as motility acquisition, the acrosomal reaction, oocyte fertilization, and early embryonic development. However, even more interestingly, certain elements of these RAS axes, such as tACE, AT2R and APN, can aid in the diagnosis and treatment of human male fertility by acting as potential biomarkers of high-quality embryos, providing added value in the pursuit of male fertility prognosis.

Even though many functions of the RAS in male fertility are clear, numerous other relevant aspects of the male reproductive system remain unknown, such as 1) whether all elements of the RAS are present in testes, sperm and seminal fluid and the functions that they exert in these tissues; 2) whether the various components play important roles in the regulation and/or formation of testes; 3) whether the various RAS elements found in the seminal fluid play roles relevant to the survival and/or transport of sperm through the female reproductive tract; and 4) whether the levels of these components could vary under different pathological conditions. A more complete knowledge of the overall function of the RAS in the male reproductive system could improve understanding of this system in sperm and in male fertility-related physiology and physiopathology. Likewise, elucidation of RAS function could support the development of new diagnostic tools or therapeutic strategies for male infertility and the establishment of useful biomarkers for the selection of optimal spermatozoa for use in assisted reproduction techniques.

## Figures and Tables

**Figure 1 ijms-21-07943-f001:**
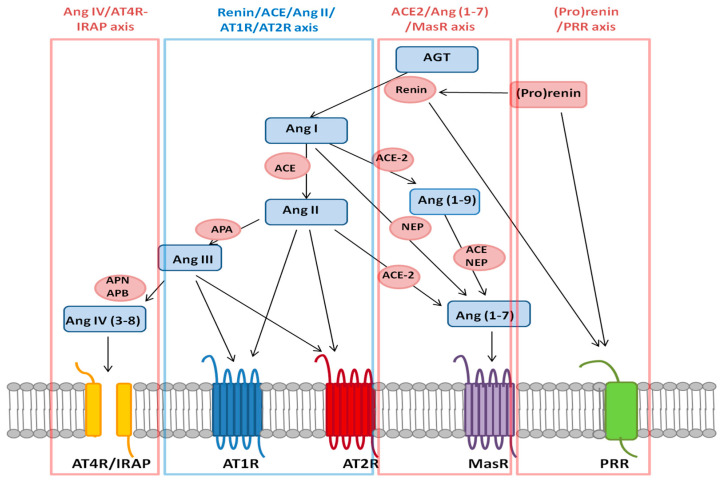
Current view of the RAS. Canonical or systemic pathway is shown in blue, while the no-canonical axes are shown in red. Peptide abbreviations: AGT, angiotensinogen; Ang I, angiotensin I; Ang II, angiotensin II; Ang III, angiotensin III; Ang IV, angiotensin (3–8) or IV; Ang (1–9), angiotensin (1–9); Ang (1–7), angiotensin (1–7). Receptor abbreviations: AT1R, Ang II receptor type 1; AT2R, Ang II type 2 receptor; AT4R/IRAP, Ang IV receptor; *MasR*, Ang receptor (1–7) or Mas receptor; PRR, (pro)renin receptor. Enzyme abbreviations: ACE, angiotensin-converting enzyme; *ACE2*, angiotensin-converting enzyme 2; NEP, neutral endopeptidase; APA, aminopeptidase A; APN, aminopeptidase N; APB, aminopeptidase B.

**Figure 2 ijms-21-07943-f002:**
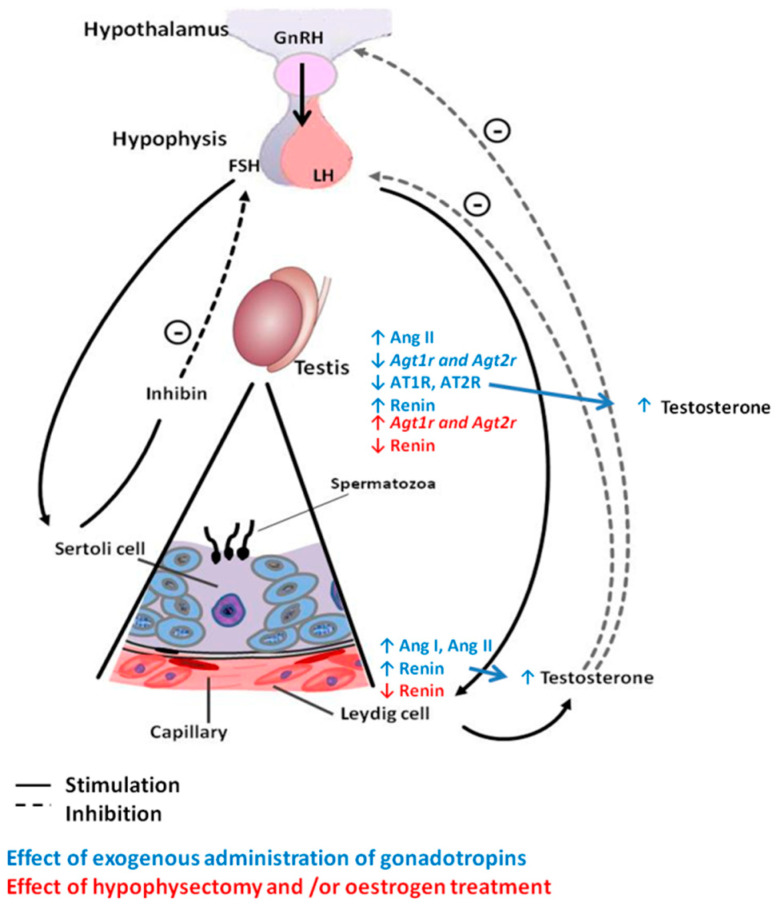
Endocrine regulation of the testicular RAS. At the testicular level, after gonadotropin stimulation, the levels of Ang II and renin increase, while the expression of the genes encoding both Ang II receptors as well as the levels of the receptors decrease. On the other hand, after hypophysectomy, the levels of renin decrease, but the mRNA expression of AT1R and AT2R increases. At the cellular level, in Leydig cells, exogenous administration of gonadotropins increases the levels of Ang I and II and renin, whereas oestrogen treatment or hypophysectomy decreases renin levels.

**Table 1 ijms-21-07943-t001:** Regulation of the testicular function by local RAS.

Axis	Component	Function	References
Renin/ACE/Ang II/ AT1R/AT2R axis	AGT	Local production of Ang II	Dzau et al. 1987, Speth et al. 1999
	ACE	Local production of Ang II	-
	Ang II	Negative regulation of testosterone production Regulation of testicular growth and/or differentiation	Dufau et al. 1989, Khanum and Dufau 1988 Schunkert et al. 1993; Leung and Sernia 2003 Hirai et al. 1998
	Ang III	Negative regulation of testosterone production	de la Chica-Rodriguez et al. 2008, Martinez-Martos et al. 2011
	AT1R	Negative regulation of spermatogenesis and testosterone production	Vinson et al. 1995 Hirai et al. 1998
	APA	Negative regulation of testosterone production	de la Chica-Rodriguez et al. 2008
*ACE2*/Ang (1–7)/ *MasR* axis	*ACE2*	Positive regulation of spermatogenesis Positive regulation of testosterone production	Reis et al. 2010 Pan et al. 2013
	Ang-(1–7)	Positive regulation of testosterone production Positive regulation of spermatogenesis	Alenina et al. 2002, Leal et al. 2009, Xu et al. 2007 Reis et al. 2010
	*MasR*	Positive regulation of spermatogenesis Regulation of testosterone production	Reis et al. 2010
	NEP	Regulation of sperm maturation and proacrosin activation	Erdos et al., 1985
Ang IV AT4R-IRAP axis	Ang IV	Negative regulation of testosterone production	de la Chica-Rodriguez et al. 2008, Martinez-Martos et al. 2011
	APN	Alteration of Sertoli cells function Negative regulation of spermatogenesis and testosterone production	Osada et al. 2001 Martinez-Martos et al. 2011
	APB	Negative regulation of testosterone production	Martinez-Martos et al. 2011
(Pro)renin/ PRR axis	(Pro)renin	Positive regulation of spermatogenesis	Mukhopadhyay et al. 1995
	Renin	Positive regulation of testosterone production	Parmentier et al. 1983

**Table 2 ijms-21-07943-t002:** Regulation of the sperm physiology by local RAS.

Axis	Component	Function	References
Renin/ACE/Ang II/ AT1R/AT2R axis	AGT	Involved in sperm-oocyte fusion	Tempfer et al. 2000
	tACE	Regulation of sperm motility, capacitation, acrosome reaction and sperm-oocyte fusion Involved in embryo quality and development	Siems et al. 1991, Yamaguchi et al. 2006, Foresta et al. 1987, Kohn et al. 1995, Hagaman et al. 1998, Krege et al. 1995, Foresta et al. 1991, Gianzo et al. 2018, Gianzo et al. 2018
	ACE3	Involved in sperm-oocyte fusion	Inoue et al. 2010
	Ang II	Regulation of sperm motility, capacitation and acrosome reaction	Foresta et al. 1991, Kohn et al. 1995, Sabeur et al. 2000, Vinson et al. 1996, Wennemuth et al. 1999
	AT1R	Regulation of sperm motility	Vinson et al. 1996
	AT2R	Regulation of sperm motility	Gianzo et al., 2016
*ACE2*/Ang (1–7)/ *MasR* axis	Ang-(1–7)	Positive regulation of sperm motility	Valdivia et al. 2020
	*MasR*	Regulation of sperm motility	Valdivia et al. 2020
	NEP	Negative regulation of sperm motility	Subiran et al. 2008
	NEP2	Negative involved in oocyte fertilization and embryo development Regulation of sperm motility	Carpentier et al. 2004 Pinto et al. 2010
Ang IV/ AT4R-IRAP axis	APN	Regulation of sperm motility and acrosome reaction Negative regulation of early embryo development	Togo and Morisawa 1997, Togo and Morisawa 2004 Subiran et al. 2008, Irazusta et al. 2004 Khatun et al. 2017
(Pro)renin/ PRR axis	PRR	Negative regulation of sperm motility	Gianzo et al. (Unpublished data)

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
