# Peer review of "Regulation of Male Fertility by the Renin-Angiotensin System"

_ijms, 2020, doi:10.3390/ijms21217943_

Round 1
Reviewer 1 Report
Review
Regulation of Male Fertility by the Renin-Angiotensin System
Marta Gianzo and Nerea Subiran
This work by Marta Gianzo and Nerea Subiran represents a comprehensive and interesting review about the connection between the Renin-Angiotensin system and male fertility. The topic is interesting and worth reviewing, since there are not so many recent reviews on this topic. Available data are analyzed with accuracy and precision.
Overall, I found the manuscript easy to read but some points need to be addressed:
- Lane 3: “Rennin” should be corrected in Renin
- Lane 4: “Subirran” should be corrected in Subiran
- Lane 112-115: the sentence should be rewritten since in the present form it doesn’t make sense
- Lane 145: “anti-hypertrophy, anti-proliferation and anti-fibrosis” cannot be properly considered physiological processes. I suggest to re-write this sentence in a most comprehensive form.
- in the introduction part, male and female data are mixed in an unclear way.
- References related to the incidence/prevalence of infertility are dated and should be substituted with most recent references (e.g Krausz C. et al., Reproduction (2015) 150 R159–R174; Deshpande P.S. & Gupta A.S. J Hum Reprod Sci (2019) 12:287-93)
- Reference 5 should be revised since some information are missing
- 1: the outlines of each rectangle should be partially transparent to allow reading of the writing below; in the figure legend please change Angiotensin 3-8 alias from Ang VI to Ang IV
- 2: the quality of the image is low; Sertoli cells should be better indicated; please correct Leyding cells to Leydig cells; in the figure legend it is unclear to what “a” and “b” are referred to; the blue and red sentences in the figure should contain the word “effect” at the end of each sentence
- Tables: please standardize the verbal forms choosing the impersonal or personal form
- Lane 193: Renin is missing in the axis and should be included
- Lane 170: references should be included at the end of the sentence
- Please correct the number of the Conclusion and perspective paragraph from 3 to 4.
- I suggest to move Tables at the end of each paragraph after all the axis were mentioned

Author Response
This work by Marta Gianzo and Nerea Subiran represents a comprehensive and interesting review about the connection between the Renin-Angiotensin system and male fertility. The topic is interesting and worth reviewing, since there are not so many recent reviews on this topic. Available data are analyzed with accuracy and precision.
We thank the reviewer for the extensive and overall positive evaluation of our manuscript. We greatly appreciate the work of the reviewer and the detailed description
Overall, I found the manuscript easy to read but some points need to be addressed:
- Lane 3: “Rennin” should be corrected in Renin, Lane 4: “Subirran” should be corrected in SubiranS
Sorry for these mistakes. We have corrected them
- Lane 112-115: the sentence should be rewritten since in the present form it doesn’t make sense. Lane 145: “anti-hypertrophy, anti-proliferation and anti-fibrosis” cannot be properly considered physiological processes. I suggest re-writing this sentence in a most comprehensive form.
Thank you for addressing this relevant point. Following the reviewer recommendations, we have re-written both sentences. Please, note that the sentences appeared in Line 125-128 and Line 164, respectively, in the new version of the manuscript.
- In the introduction part, male and female data are mixed in an unclear way.
As reviewer has recommended, we have re-written the introduction part, in order to clarify this paragraph
- References related to the incidence/prevalence of infertility are dated and should be substituted with most recent references (e.g Krausz C. et al., Reproduction (2015) 150 R159–R174; Deshpande P.S. & Gupta A.S. J Hum Reprod Sci (2019) 12:287-93)
Reference 5 should be revised since some information are missing
Thank you very much for highlight this point, we have changed the first reference with most recent reference and corrected the Reference 5 information.
- 1: the outlines of each rectangle should be partially transparent to allow reading of the writing below; in the figure legend please change Angiotensin 3-8 alias from Ang VI to Ang IV
- 2: the quality of the image is low; Sertoli cells should be better indicated; please correct Leyding cells to Leydig cells; in the figure legend it is unclear to what “a” and “b” are referred to; the blue and red sentences in the figure should contain the word “effect” at the end of each sentence
Both figures and figure legends have changed following the recommendations.
- Tables: please standardize the verbal forms choosing the impersonal or personal form I suggest to move Tables at the end of each paragraph after all the axis were mentioned
Thanks for this important point. We have standardized and moved tables at the end of each paragraph
- Lane 193: Renin is missing in the axis and should be included
Lane 170: references should be included at the end of the sentence
Please correct the number of the Conclusion and perspective paragraph from 3 to 4
Sorry for these mistakes. We have corrected all of them. In the new version of the manuscript they correspond to the Line 199, Line 184 and Line 453

Reviewer 2 Report
Dear authors, I read carefully your review and I believe that is well written and suitable for publication on IJMS.
I would like just to underline a couple of minor thinks that could be improved. In introduction you could underline not only separate male and female factors but even mix factors when both males/females are affected. Furthermore when talking about causes of male infertility you could organize better this section and underline as 30% is idiopathic and other known causes are varicocele, genetic and hormonals. Finally on line 170 you forgot the reference.
Author Response
Dear authors, I read carefully your review and I believe that is well written and suitable for publication on IJMS.
We thank the reviewer for the extensive and overall positive evaluation of our manuscript. We greatly appreciate the work of the reviewer and the detailed description
I would like just to underline a couple of minor thinks that could be improved.
In introduction you could underline not only separate male and female factors but even mix factors when both males/females are affected. Furthermore when talking about causes of male infertility you could organize better this section and underline as 30% is idiopathic and other known causes are varicocele, genetic and hormonals.
As reviewer has recommended, we have re-written the introduction part, in order to clarify this paragraph
Finally on line 170 you forgot the reference.
Thank you very much for highlight this point, we have included references. In the new version of the manuscript it corresponds to Line 184.